# Potential Biocontrol Agents of Corn Tar Spot Disease Isolated from Overwintered *Phyllachora maydis* Stromata

**DOI:** 10.3390/microorganisms11061550

**Published:** 2023-06-10

**Authors:** Eric T. Johnson, Patrick F. Dowd, José Luis Ramirez, Robert W. Behle

**Affiliations:** Crop Bioprotection Research Unit, National Center for Agricultural Utilization Research, Agricultural Research Service, United States Department of Agriculture, 1815 N University Street, Peoria, IL 61604, USA; patrick.dowd@usda.gov (P.F.D.); jose.ramirez2@usda.gov (J.L.R.); robert.behle@usda.gov (R.W.B.)

**Keywords:** *Phyllachora maydis*, tar spot disease, corn pathogen, biological control, biofungicide

## Abstract

Tar spot disease in corn, caused by *Phyllachora maydis*, can reduce grain yield by limiting the total photosynthetic area in leaves. Stromata of *P. maydis* are long-term survival structures that can germinate and release spores in a gelatinous matrix in the spring, which are thought to serve as inoculum in newly planted fields. In this study, overwintered stromata in corn leaves were collected in Central Illinois, surface sterilized, and caged on water agar medium. Fungi and bacteria were collected from the surface of stromata that did not germinate and showed microbial growth. Twenty-two *Alternaria* isolates and three *Cladosporium* isolates were collected. Eighteen bacteria, most frequently *Pseudomonas* and *Pantoea* species, were also isolated. Spores of *Alternaria*, *Cladosporium*, and *Gliocladium catenulatum* (formulated as a commercial biofungicide) reduced the number of stromata that germinated compared to control untreated stromata. These data suggest that fungi collected from overwintered tar spot stromata can serve as biological control organisms against tar spot disease.

## 1. Introduction

Corn is one of the most important grain crops in the world, but production is limited by a variety of pests and diseases [1]. One devastating disease is caused by *Phyllachora maydis* and is commonly referred to as tar spot because of its characteristic black, shiny, raised leaf spots which generally range from 2–4 mm in diameter. Infection by this pathogenic fungus can result in significant yield losses, and even death of plants if infection occurs early in a susceptible variety [2]. Tar spot disease has been endemic in much of Central and South America for several decades [2]. It was first reported in two U.S. “corn belt” states, Illinois and Indiana, in 2015 [3], and since then there have been major outbreaks in several regions in both 2018 and 2021 [4].

Management strategies for tar spot disease include use of resistant varieties and application of fungicides [2]. However, under high inoculum pressure, significant yield losses can still occur, even with the use of resistant varieties [5]. Fungicide application has not been very effective because once symptoms are noticed, further infection becomes difficult to control [6].

The black structures characteristic of tar spot disease are long term survival structures called stromata, which are similar to the sclerotia produced by the vegetable pathogen *Sclerotinia sclerotiorum* and other fungi [7]. These stromata are the overwintering structures for tar spot, and the source of initial inoculum. However, reports of spore production and percent germination by overwintered stromata vary considerably [8], suggesting the presence of natural biocontrol organisms. Mycoparasites may contribute to reduced survival of stromata, as reported for other stromata or sclerotia-like structures in several fungal species [7,9] For example, several species of biocontrol organisms have been isolated from the sclerotia of *S. sclerotiorum*, some of which have been commercialized as biological control products [10,11]. Mycoparasites of stromata of *Coccodiella miconiae*, a relative of *P. maydis*, included fungi in the genera *Cladosporium*, *Cornespora*. and *Sagenomella* [12]. To investigate the potential for biocontrol of tar spot, stromata from overwintered corn leaves collected in Illinois were examined for the presence of mycoparasites or other biocontrol agents. This study identifies several bacterial and fungal species infecting tar spot stromata and describes bacterial and fungal species with potential to serve as biocontrol agents against tar spot disease of corn.

## 2. Materials and Methods

### 2.1. Isolation of Organisms

Several overwintered corn leaves of inbred GE440 (planted from seed increased from plants originally obtained from the USDA-ARS Plant Introduction Center) were collected in late April 2022 from a 2021-planted research plot. This site in Peoria, IL, has had continuous corn production for several years and is thus more ecologically stable and likely to yield biocontrol agents for tar spot than commercial fields where crop rotation typically occurs. Stromata were cut from leaves along with a small leaf piece “handle”. Approximately 50 stromata were surface sterilized with 70% ethanol and then blotted dry as described previously [8]. Stromata were placed in Petri dishes with tight fitting lids (Falcon^R^ 351006, Corning Inc., Corning, NY, USA) containing 5 mL of 3% water agar to induce rehydration and stimulate growth of mycoparasites; in some cases, a few µL of sterile water was added to help with rehydration. Plates were held at 25–27 °C. Stromata were examined daily for outgrowth of organisms that were not visually the same as any seen from the attached leaf material, and organisms noted were photographed with cameras equipped with macro lenses. Microorganisms were isolated using two different methods. In the first method, fungal mycelium growing out of a single stroma was transferred to a potato dextrose agar (PDA, Difco potato dextrose broth, Becton Dickinson Company, Sparks, MD, USA, with bacto agar at 20 g/L) plate using a sterile metal probe. A few days later, bacteria were observed growing together with the fungus. The bacteria were transferred to a new PDA plate, whereas the bacterium-contaminated fungus was transferred to a PDA plus 0.01% chloramphenicol plate. In the second method, petroleum jelly was used to stick leaves with stromata to the inside of the top lid over a 3% water agar plate. After 2–3 days, microorganisms growing on the surface of the agar were transferred to nutrient media plates (Luria broth (LB) or tryptone glucose yeast extract (TGY) for bacteria or PDA for fungi). LB agar plates consisted of tryptone (10 g/L), sodium chloride (10 g/L), yeast extract (5 g/L), and bacto agar at 15 g/L. TGY agar plates consisted of tryptone (5 g/L), yeast extract (5 g/L), K_2_HPO_4_ (1 g/L), and glucose (1 g/L); the pH of the final mixture was adjusted to 7.0 and bacto agar was added at 15 g/L before autoclaving.

### 2.2. Identification of Organisms

Genomic DNA was isolated from fungi as previously described [13] with a modification: a small fragment of fungal mycelia from the culture plate was pulverized with a small amount of 800 µm silica beads using a Minibead Beater (Biospec Products, Bartlesville, OK, USA). Genomic DNA was isolated from bacteria as described in [14] with several modifications. A 1 mL aliquot of bacterial overnight culture was centrifuged at 16,000× *g*; the bacterial pellet was first suspended in 480 µL of 50 mM EDTA and 120 µL of 5 mg/mL lysozyme was then added. The suspended pellet was incubated at 37 °C for 30–60 min and the suspension was centrifuged at 16,000× *g* for 2 min. The supernatant was moved to a new tube and 600 µL of nuclei lysis solution was added. The lysate was incubated at 80 °C for 5 min. After cooling to room temperature, 12 µg of ribonuclease A was added to the lysate and the mixture was incubated at 37 °C for 15–60 min. Then, 250 µL of 5 M NaCl was added to the lysate and the mixture was vortexed for 20 s. The mixture was kept on ice for 5 min and then centrifuged at 16,000× *g* for 3 min. The method was continued as previously described [14]. Genomic DNA was stored at −20 °C until processing. PCR amplification of various gene products from genomic DNA was performed using Platinum™ SuperFi™ II DNA polymerase (Thermo Fisher Scientific, Waltham, MA, USA) according to the manufacturer’s instructions; primers used for PCR product amplification are listed in Appendix A. PCR products were sequenced using the BigDye Terminator Cycle Sequencing Kit (Version 3.1, Applied Biosystems, Foster City, CA, USA) and BLAST analysis (National Center for Biotechnology Information (NCBI) was used to determine potential identities [15]. For bacteria, conserved gene sequences targeting the *16S rDNA* gene were amplified by PCR, sequenced, and utilized for identification of each organism. In some cases of bacterial identification, a portion of the *gyrB* gene was also amplified from genomic DNA because the *16S rDNA* gene sequence was not suitable to identify the bacterium to the species level. The threshold for assigning a bacterial sequence to the species level was 98% or greater similarity of the PCR product sequence with a GenBank accession from the nucleotide collection or the whole genome shotgun contig database. For fungi, conserved gene sequences targeting the *rDNA* gene were amplified from genomic DNA using PCR and sequenced; these PCR products could not identify the fungi to the species level. Therefore, a portion of the actin gene was amplified by PCR in the *Cladosporium* fungi [16,17], whereas portions of three genes (RNA polymerase second largest subunit, *rpb2*; *Alternaria* major allergen, *Alt-a1*; glyceraldehyde 3-phosphate dehydrogenase, *gapdh*) were amplified via PCR in the *Alternaria* fungi. These *Alternaria* genes were sufficient to distinguish new species in a recent paper [18]. Phylogenetic analysis (see below) was used to identify *Cladosporium* and *Alternaria* fungi to the species level. All the sequencing results were supplied in the Appendix A.

### 2.3. Repression of Stromata Germination by Representative Fungal Potential Biocontrol Agents

Corn leaves with tar spot stromata were collected from a field in Peoria County, IL, in late July 2022. Leaves were rubbed gently while held under flowing deionized water to dislodge debris and blotted dry with Wipeall^R^ L40 wipes (Kimberly ClarkRoswell, GA, USA). Leaves containing approximately 50 to 100 *P*. *maydis* stromata in a 100 cm^2^ square were trimmed with the midvein central so they would fit snugly and flat in 100 × 15 mm Petri dishes containing 25 mL of 3% water agar. Plates were sealed with 3 M micropore™ surgical tape (3 M Company, ST. Paul, MN, USA) and allowed to incubate at room temperature overnight. The experiment was set up with three treatments, including two representatives of the most common genera of potential biocontrol fungi isolated from overwintered stromata (*Alternaria alternata/arborescens* and *Cladosporium rectoides*), and the commercial biofungicide LALstop G46wg, which contains *Gliocladium catenulatum* strain J1446, obtained from Lallemand Specialties (Milwaukee, WI, USA). Spores were obtained from fresh cultures of potential biocontrol fungi that were grown on S-medium [19]. Solutions of spores from the candidate biocontrol fungi were diluted to 10 colony forming units per µL in 0.01% Triton X 100 (Sigma Chemical, St. Louis, MO, USA). The commercial biofungicide LALstop was also diluted to 10 CFU per µL to match that for the candidate biocontrol fungi from the stromata, which was comparable to the recommended application rate. The next day, stromata that had not germinated were treated with 1 µL per mm diameter of control or spore solution, with 10 stromata treated on each side of the leaf midrib with either control (Triton X 100 solution alone) or spore solution. Plates were allowed to incubate at room temperature for 4 days, and then the number of treated germinated stromata was determined. Each treatment was replicated 3 times.

### 2.4. Phylogenetic Analysis

Partial gene sequences from *Alternaria* (*Alt-a1, gapdh,* and *rpb2* genes) and *Cladosporium* (actin gene) ex-type species were retrieved from NCBI. Sequences of *Cladosporium* ex-type strains, ex-epitype strains, and the strains from this study were aligned (using the Muscle alignment tool) and phylogenetic tree constructed using MEGA 11 [20]. Sequences of *Alternaria* ex-type strains and the strains from this study were aligned (using the “very accurate” algorithm in the classical sequence analysis menu) in CLC Genomics Workbench Version 22.0.2 (Qiagen, Valencia, CA, USA). Each *Alternaria* gene was aligned individually and trimmed at the edges; nucleotides within the gene that had low consensus among all the sequences were removed from the alignment. A multi-locus alignment of the three genes was made using the “join alignments” function. The alignment was imported into MEGA 11 and the phylogenetic tree constructed.

### 2.5. Statistical Analysis

Significant differences in frequency of tar spot stromata germination between control treated stromata and biological control organism treated stromata were determined using weighted Chi Square analysis with the SAS Version 7.1 Proc Freq (SAS Institute, Cary, NC, USA).

## 3. Results

No germinating stromata were observed in any collected overwintered tar spot infected material. However, several different types of organisms were observed growing out from stromata and not the leaf “handle” (Figure 1). We recovered 43 isolates of bacteria and fungi from approximately 50 of the *P. maydis* stromata (Table 1). Bacteria isolates included four species of *Pantoea*, including *P*. *agglomerans* (3 isolates), as well as two isolates of *Priestia* (formerly known as *Bacillus*) *megaterium*. Other bacteria isolated included *Curtobacterium faccumfaciens*, and multiple species of *Pseudomonas*, including *P. graminis*, *P. prosekii*, and *P*. *quercus*. One *Pseudomonas* isolate is either *P. fluorescens* or *P. shahriarae.*

*Alternaria* was the most common fungus isolated from the stromata (Table 1). The *Alternaria* fungi could not be identified to the species level using the amplified region of the *rDNA* gene. PCR amplification of three gene fragments and subsequent phylogenetic analysis of the multi-locus alignment indicated that 6 isolates were *A. ovoidea* and 15 were possibly *A. alternata* or *A. arborescens* (Appendix A). Three putative *Cladosporium* strains were isolated. The phylogenetic analysis (Appendix A) indicated the three *Cladosporium* isolates were *C. crousii*, *C. rectoides*, and *C. subuliforme*.

The number of germinating stromata as indicated by exudate presence was significantly reduced by spore suspensions of two representative fungal species isolated from tar spot stromata compared to control applications (Table 2). However, none of these fungi were as effective as the commercially available *G. catenulatum*.

## 4. Discussion

### 4.1. Biocontrol Potential of Bacteria Isolated from Tar Spot Stromata

Of the 18 bacteria isolated from tar spot stromata, only *P. megaterium* has been reported from the corn microbiome [21]. Some bacterial species that we isolated from tar spot stromata have previously been reported as biocontrol agents or have properties of biocontrol agents. Due to limited available leaf material with the needed density of stromata, bioassay evaluation of representative bacteria was not performed. Although the *Curtobacterium* species we isolated from tar spot stromata has not previously been reported as a biocontrol agent, *Curtobacterium* spp. can be present in plants as pathogens or endophytes. An undescribed species of *Curtobacterium* was chitinolytic and had potential as a biocontrol agent [22]. *Pantoea* (formerly *Enterobacter*) *agglomerans* is reported as a biocontrol for many fungal pathogens such as *Botrytis cinera, Pythium* sp., and *Sclerotinia sclerotiorum* in several fruit, vegetable, and row crops [23]. Although many strains of *Pantoea ananatis* are plant pathogens, one has been reported as mycoparasite of wheat leaf rust, *Puccinia graminis* [24]. A strain of *P. anthophola* antagonized the growth of *Ralstonia solanacearum* [25]. A non-nitrogen-fixing strain of *P. eucalypti* promotes the growth of the pine *Pinus massoniana* [26], which could be due to inhibition of pathogens. Other species of *Pantoea* have also been reported as biocontrol agents [27].

*Priestia megaterium* has been reported to control septoria tritici blotch [28] of wheat caused by ascomycete fungus, *Mycophaerella graminicola*, multiple species of mycotoxigenic fungi [29], and other fungal pathogens in diverse crops [23]. Other *Priestia* species have also been reported as biocontrol agents [30]. *Pseudomonas graminis* has been reported as an antagonist of the fire blight causal organism *Erwinia amylovora* [31]. *Pseudomonas prosekii* inhibits plant pathogenic strains of *P. fluorescens* and *P. viridiflava* [32]. *Pseudomonas fluorescens* has been widely reported as a biological control agent effective against both bacterial and fungal plant pathogens, and different strains have been commercialized in several instances [33]. For example, pathogen inhibition by *P*. *fluorescens* has been reported for *Venturia inaequalis* in apple, *Macrophomina phaseolina* in mung bean, *Fusarium moniliforme* in cauliflower, *Xanthomonas campestris* pv. *malvacearum* in cotton, *Puccinia arachidis* in peanut, *Magnaporthe grisea* in rice, *Rhizoctonia solani* in tomato, and *Helminthosporium sativum* in wheat [33]. No biological control properties have been described for *Pseudomonas shahriarae*, which was isolated from the rhizosphere of wheat growing in Iran [34].

### 4.2. Biocontrol Potential of Fungi Isolated from Tar Spot Stromata

The commercial fungus formulation of *G*. *catenulatum* was very effective at inhibiting germination of *P maydis* stromata in corn leaves. *Alternaria alternata/arborescens* strain 11C + F, which was isolated from an overwintered *P. maydis* stroma, reduced germination of stromata in growing season leaves at statistically significant levels compared to the control leaves but was not as effective at reducing germination rates as *G. catenulatum*. *Cladosporium rectoides* strain 10RDF reduced germination rates of stromata in growing season leaves but the germination rate was not lower than the germination rate in the control leaves at a statistically significant level (*p* = 0.0653). Of the fungi isolated from tar spot stromata, only *A. alternata* has been reported from the corn microbiome [21]. Because different taxonomic approaches can identify the same fungus as two different species [35], it is challenging to compare past reports of fungal biocontrol agents that were predominately based on morphological identifications with isolated species identified using molecular methods in the present study. However, several species in the two genera of fungi we isolated from tar spot stromata have been reported as biocontrol agents. Eight species of *Alternaria* are reported as biocontrol agents [36], one of which, *A. alternata*, we identified in the present study. *A. alternata* is reported as a biocontrol agent for tan spot of wheat caused by *Pyrenophora tritici-repentis* [37], verticillium wilt of cotton caused by *Verticillium dahlia* [38], and white mold of bean caused by *Sclerotinia sclerotiorum* [39]. *Alternaria ovoidea*, which we isolated in the present study, has been reported as a saprophyte of the grass *Dactylis glomerata* [18]. Based on the saprophytic capabilities of some biocontrol agents of sclerotia described previously, it would not be surprising that *A. ovoidea* could also function as a mycoparasite. The other species of *Alternaria* reported previously as biocontrol agents have been used on banana, cyclamen, cotton, grape onion, roses, and strawberry, primarily against *Botrytis* and *Verticillium* spp. pathogens [36].

One species of *Cladosporium*, C. *subuliforme*, which we isolated from tar spot stromata in the present study, has previously been reported as a biocontrol agent. *C. subuliforme* from the rice phylloplane inhibited the growth of four rice fungal pathogens in dual culture plate assays [40]. Five other species of *Cladosporium* have been reported as biocontrol agents [36], although none of the species were the same as those identified in the present study. These additional species of *Cladosporium* have been used as biocontrol agents on chrysanthemum, cotton, guava, and grape for control of *Colletotrichum, Eutypa, Puccinia*, and *Verticillium* spp. pathogens [36].

### 4.3. Potential Role of Biocontrol in Tar Spot Management

The disease cycle of the tar spot pathogen is not well understood, although it is thought that the overwintered stromata are the source of initial infective material each growing season, and that the disease can spread through a corn field when spores are released by germinating stromata [2]. Plant resistance has been suggested and utilized to help reduce tar spot disease [2]. However, under certain conditions, even resistant varieties can be severely infected by the tar spot pathogen [5]. Although some fungicides are labeled for tar spot disease control (e.g., Veltyma), only spores and mycelia are listed as targets, not stromata. Stromata from fields treated with fungicide labeled for tar spot will still sporulate a few weeks after treatment (authors, personal observation). However, there are commercial products labeled to control species of sclerotia forming fungi, including *Coniothyrium minitans* (Contans WG), *G. castenulatum* (LALstopG46), and *Trichoderma harzianum* (Trianum-G); it should be noted that sclerotia are analogous to *P*. *maydis* stromata. Previous reports indicated these species can control several species of sclerotia-forming fungi, although the tar spot organism is not mentioned as one of them; *G*. *castenulatum* and *T*. *harzianum* are described as “ecologically facultative” [7] and thus can also survive as saprophytes and potentially persist to exert long-term control. The commercial strain of *G. castanulatum*, which is labeled for use in many crops, including corn, has saprophytic capabilities [41], and this strain was able to prevent germination of tar spot stromata as indicated in the present study. *Trichoderma harzianum* has been used to control foliar disease in corn [42].

The present study indicates that overwintered tar spot stromata can also be a source of useful biological control organisms, as two representative isolates significantly reduced the percentage of stromata germination compared to the control treatment. Stromatal isolates may have advantages over existing commercial materials such as the ability to grow and infect tar spot stromata under cooler conditions, making them more suitable for applications in the fall season to reduce the viability of overwintering stromata. However, the present study also suggests that overwintering fungi can be applied during the growing season and provide some control of stromata to augment fungicides that target spores and mycelia of the tar spot pathogen. Infected nongerminating stromata have also been collected during the growing season at different locations in 2022 and were common in late season stromata collected from the site investigated in the present study indicating growing season applications could also be used to reduce overwintering survival of tar spot stromata. However, further study is needed to identify the most effective organisms/strains under proposed field application conditions and determine whether undesirable traits, such as plant pathogenicity or other genes that produce harmful proteins, are present.

## 5. Conclusions

The present study suggests that survival of overwintering tar spot stromata can be greatly reduced by both bacterial and fungal mycoparasites. Some of the species of organisms that we recovered from stromata have previously been reported to be biocontrol agents. Three example microorganisms that we tested significantly reduced the percent sporulation of growing season stromata, suggesting that mycoparasites can be used as part of an integrated management plan for both early season preventative and growing season control measures for tar spot disease in corn.

## Figures and Tables

**Figure 1 microorganisms-11-01550-f001:**
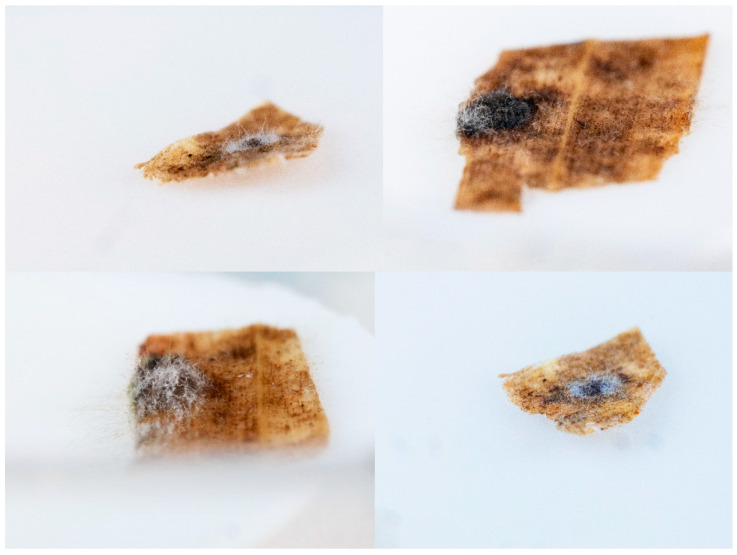
Several examples of a possible biological control organism emerging from *P. maydis* overwintered stromata.

**Table 1 microorganisms-11-01550-t001:** Identities of organisms recovered from overwintered tar spot stromata.

Organisms Directly Removed from Stromata	Number of Individual Organisms	Biological Control Properties Reported
Fungi		
*Alternaria*	1	
*A. alternata/A. arborescens*	11	Yes *
*Alternaria ovoidea*	6	Yes
*Cladosporium rectoides*	1	Yes *
*Cladosporium subuliforme*	1	Yes
Bacteria		
*Curtobacterium flaccumfaciens*	2	
*Pantoea agglomerans*	3	Yes
*Pantoea ananatis*	1	Yes
*Pantoea anthophila*	1	Yes
*Pantoea eucalypti*	2	Yes
*Priestia megaterium*	2	Yes
*Pseudomonas graminis*	3	Yes
*Pseudomonas prosekii*	1	Yes
*Pseudomonas quercus*	1	
**Organisms that fell onto water agar plates from stromata**		
Fungi		
*A. alternata/A. arborescens*	4	Yes
*Cladosporium crousii*	1	
Bacteria		
*Priestia flexa*	1	
*Pseudomonas fluorescens* or *shahriarae*	1	

* Reported in this study for the first time; isolates *A. alternata/A. arborescens* 11C + F and *Cladosporium rectoides* 10RDF exhibited biological control properties.

**Table 2 microorganisms-11-01550-t002:** Inhibition of tar spot stromata germination by putative biocontrol fungi.

Organism	Source	% Germination Control	% Germination with Organism	Χ^2^ Value	*p* Value
*Gliocladium* *catenulatum*	Commercial	83.3	10.0	32.4107	<0.0001
*Alternaria* *alternata/* *arborescens 11C + F*	Present study	66.7	36.7	5.4060	0.0201
*Cladosporium* *rectoides 10RDF*	Present study	50.0	26.7	3.4548	0.0653

% germination values are based on three replicates of 10 stromata for each treatment per replicate, with each replicate having the control and fungal treatments on opposites sides of the same leaf.

## Data Availability

All data generated in this study are available in the Tables and Appendix A.

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
