# Peer review of "Potential Biocontrol Agents of Corn Tar Spot Disease Isolated from Overwintered Phyllachora maydis Stromata"

_microorganisms, 2023, doi:10.3390/microorganisms11061550_

Round 1

Reviewer 1 Report

This manuscript was well written and the length was appropriate for the experimental design, results, and discussion. There are just a few items that must be addressed and then some suggestions that I think would make a slightly stronger paper.

The formatting of the citations in the text and the lit cited section do not follow the most recent publications in Microorganisms.  The journal uses the numbering system for in-text citations and the references in the literature cited occur in chronological order following the order of first citation in the text. 

Line 36 Brandt 2022 is cited but this reference is not in the literature cited section.

Line 111-112: I think it is important to describe what the threshold was for assigning a sequence to genbank reference that was "clearly the same species".  Is this greater the 98% similarity, 95% similarity?  This is the only instance in the methods where I could not repeat what the authors did in the their study.  Overall, the methods section was nicely written.

Lines 184-190.  I had a challenging time understanding precisely what the actual results were as they were written in the text.  I assumed that the first % number given was the percent inhibition of stromata germination by G. catenulatum and the following number was that of the of the potential fungal biological control agents. Because these tests of germination inhibition were actually very important to the study, I think that they should be presented in their own table or figure so the results can be readily found by the reader and play a more prominent visual role in the paper to complement the conceptual role they play in the paper. 

Discussion

Section 4.1.  I think it would be nice if the authors reminded the reader that bacteria were not evaluated experimentally for the suppression of stromata germination somewhere in this section.  I don't feel that this a something required but it would transparently frame this part of the Discussion section. I had to reread the methods to make sure that the bacteria were not evaluated because of the treatment in section 4.1.  I do not have a problem with the treatment as it is calling the reader's attention to the bacteria associated with tar spot and there potential for biologically antagonistic interactions with a plant pathogen. It is just a bit confusing to read this topic first in the Discussion when the suppressive impacts of bacteria were not tested like the fungi were. 

Author Response

Reviewer 1

This manuscript was well written and the length was appropriate for the experimental design, results, and discussion. There are just a few items that must be addressed and then some suggestions that I think would make a slightly stronger paper.

The formatting of the citations in the text and the lit cited section do not follow the most recent publications in Microorganisms.  The journal uses the numbering system for in-text citations and the references in the literature cited occur in chronological order following the order of first citation in the text. 

AUTHORS RESPONSE: MICROORGANISMS ALLOWS “FREE FORMAT SUBMISSION” OF MANUSCRIPTS.  IN ADDITION, THE INSTRUCTIONS FOR AUTHORS STATES:

YOUR REFERENCES MAY BE IN ANY STYLE, PROVIDED THAT YOU USE THE CONSISTENT FORMATTING THROUGHOUT. IT IS ESSENTIAL TO INCLUDE AUTHOR(S) NAME(S), JOURNAL OR BOOK TITLE, ARTICLE OR CHAPTER TITLE (WHERE REQUIRED), YEAR OF PUBLICATION, VOLUME AND ISSUE (WHERE APPROPRIATE) AND PAGINATION. DOI NUMBERS (DIGITAL OBJECT IDENTIFIER) ARE NOT MANDATORY BUT HIGHLY ENCOURAGED. THE BIBLIOGRAPHY SOFTWARE PACKAGE ENDNOTE, ZOTERO, MENDELEY, REFERENCE MANAGER ARE RECOMMENDED.

Line 36 Brandt 2022 is cited but this reference is not in the literature cited section.

AUTHORS RESPONSE: THE BRANDT 2022 REFERENCE WAS ACCIDENTALLY PLACED INTO THE BOLAND AND INGLIS 1989 REFERENCE.  THE BRANDT 2022 REFERENCE HAS BEEN ADDED TO THE MANUSCRIPT.

Line 111-112: I think it is important to describe what the threshold was for assigning a sequence to genbank reference that was "clearly the same species".  Is this greater the 98% similarity, 95% similarity?  This is the only instance in the methods where I could not repeat what the authors did in the their study.  Overall, the methods section was nicely written.

AUTHORS RESPONSE:

THIS SENTENCE:

THE THRESHOLD FOR ASSIGNING A BACTERIAL SEQUENCE TO THE SPECIES LEVEL WAS 98% OR GREATER SIMILARITY OF THE PCR PRODUCT SEQUENCE WITH A GENBANK ACCESSION FROM THE NUCLEOTIDE COLLECTION OR THE WHOLE GENOME SHOTGUN CONTIG DATABASE.

REPLACED THE FOLLOWING SENTENCE IN SECTION 2.2

BACTERIA WERE IDENTIFIED TO THE SPECIES LEVEL WHEN ONE OR SEVERAL OF THE TOP MATCHING BLAST HITS (ALIGNED AGAINST SEQUENCES FROM THE NUCLEOTIDE COLLECTION OR THE WHOLE GENOME SHOTGUN CONTIG DATABASE) TO THE PCR PRODUCT SEQUENCE WERE CLEARLY THE SAME SPECIES.

Lines 184-190.  I had a challenging time understanding precisely what the actual results were as they were written in the text.  I assumed that the first % number given was the percent inhibition of stromata germination by G. catenulatum and the following number was that of the of the potential fungal biological control agents. Because these tests of germination inhibition were actually very important to the study, I think that they should be presented in their own table or figure so the results can be readily found by the reader and play a more prominent visual role in the paper to complement the conceptual role they play in THE PAPER. 

AUTHORS RESPONSE: AN ADDITIONAL TABLE WAS ADDED TO THE MANUSCRIPT AND THE RESULTS SECTION EDITED ACCORDINGLY.

Discussion

Section 4.1.  I think it would be nice if the authors reminded the reader that bacteria were not evaluated experimentally for the suppression of stromata germination somewhere in this section.  I don't feel that this a something required but it would transparently frame this part of the Discussion section. I had to reread the methods to make sure that the bacteria were not evaluated because of the treatment in section 4.1.  I do not have a problem with the treatment as it is calling the reader's attention to the bacteria associated with tar spot and there potential for biologically antagonistic interactions with a plant pathogen. It is just a bit confusing to read this topic first in the Discussion when the suppressive impacts of bacteria were not tested like the fungi were. 

AUTHORS RESPONSE: THE FOLLOWING SENTENCE WAS ADDED TO SECTION 4.1 AS RECOMMENDED.

Due to limited amount of leaf material with the needed density of stromata, bioassay evaluation of representative bacteria was not performed.

Reviewer 2 Report

The approach of isolating biocontrol microbes from overwintering stromata which are not germinating is very interesting. However, the manuscript is not detailed enough in presenting the methods of identification. For example, in line 105/106, the primers used for the amplification of 16SrRNA genes of bacteria should be presented. Also, the presentation of detailed results about the biocontrol activities of the isolates (bacteria and fungi) against P. maydis is missing.  

Author Response

Reviewer 2

The approach of isolating biocontrol microbes from overwintering stromata which are not germinating is very interesting. However, the manuscript is not detailed enough in presenting the methods of identification. For example, in line 105/106, the primers used for the amplification of 16SrRNA genes of bacteria should be presented. Also, the presentation of detailed results about the biocontrol activities of the isolates (bacteria and fungi) against P. maydis is missing.  

AUTHORS RESPONSE: WE THINK THE REVIEWER MISSED READING ABOUT PCR PRIMERS IN LINES 102-103 (ORIGINAL SUBMISSION):

PRIMERS USED FOR PCR PRODUCT AMPLIFICATION ARE LISTED IN SUPPLEMENTARY FILE 1.

DO TO LIMITED AVAILABLE LEAF MATERIAL WITH THE NEEDED DENSITY OF STROMATA, ONLY A FEW BIOASSAYS USING TWO OF THE FUNGAL ISOLATES COULD BE PERFORMED.  THE RESULTS OF THE BIOCONTROL ACTIVITIES ARE NOW NOTED IN TABLE 2.  IT SHOULD BE NOTED THAT REVIEWER 1 FOUND OUR METHODS SECTION “NICELY WRITTEN” EXCEPT FOR ONE DETAIL.